# Neural Classification of Compost Maturity by Means of the *Self-Organising Feature Map* Artificial Neural Network and *Learning Vector Quantization* Algorithm

**DOI:** 10.3390/ijerph16183294

**Published:** 2019-09-07

**Authors:** Piotr Boniecki, Małgorzata Idzior-Haufa, Agnieszka A. Pilarska, Krzysztof Pilarski, Alicja Kolasa-Wiecek

**Affiliations:** 1Institute of Biosystems Engineering, Poznań University of Life Sciences, Wojska Polskiego 50, 60-637 Poznań, Poland; bonie@up.poznan.pl (P.B.); pilarski@up.poznan.pl (K.P.); 2Department of Gerodontology and Oral Pathology, Poznań University of Medical Sciences, Bukowska 70, 60-812 Poznań, Poland; midziorhaufa@ump.edu.pl; 3Institute of Food Technology of Plant Origin, Poznań University of Life Sciences, Wojska Polskiego 31, 60-637 Poznań, Poland; 4Institute of Technical Sciences, Opole University, ul. Dmowskiego 7-9, 45-365 Opole, Poland; akolasa@uni.opole.pl

**Keywords:** non-parametric classification, SOFM neural network, *LVQ* algorithm

## Abstract

*Self-Organising Feature Map (SOFM)* neural models and the *Learning Vector Quantization (LVQ)* algorithm were used to produce a classifier identifying the quality classes of compost, according to the degree of its maturation within a period of time recorded in digital images. Digital images of compost at different stages of maturation were taken in a laboratory. They were used to generate an *SOFM* neural topological map with centres of concentration of the classified cases. The radial neurons on the map were adequately labelled to represent five suggested quality classes describing the degree of maturation of the composted organic matter. This enabled the creation of a neural separator classifying the degree of compost maturation based on easily accessible graphic information encoded in the digital images. The research resulted in the development of original software for quick and easy assessment of compost maturity. The generated SOFM neural model was the kernel of the constructed IT system.

## 1. Introduction

Due to clients’ increasing expectations concerning changes in the quality of agricultural products, it is necessary to use modern identification methods, such as computer-assisted data analysis, including inventive neural modelling systems. These systems simulate complex processes of agricultural production in its broad sense [1]. The superiority of neural models over traditional data processing systems mainly results from the fact that they try to reproduce the work of the human brain [2]. In *Artificial Neural Networks (ANN),* traditional programing was replaced by the training process, whereas series data processing was replaced by parallel calculations [3,4]. In consequence, the acquisition of information at the output of the generated neural model depends on the configuration and strength of connections (so-called synaptic balances) between cooperating elements (neurons), which are nodes of the *ANN* graphical structures. The dynamic development of data analysis methods and data mining techniques increased interest in self-organising *ANN* learning without a teacher [5,6,7]. The tuning-in of balances, taking place in these networks during the training process, is done through the ‘no pattern’ technique, which uses dedicated optimizing algorithms. *Self-Organising Feature Map (SOFM)* neural models, proposed by Kohonen, are some of the most commonly used types of these networks [8]. A key characteristic of *SOFM* is its ability to provide a topology preserving the mapping of a high dimensional input (feature) space onto a low dimension grid [8]. *SOFM* is particularly suitable for data visualization and analysis because it conveniently facilitates the use of unique human insight, as it visualizes the relationships between the input vectors in high dimensional space from a two-dimensional display, which is otherwise impossible. The ability to classify objects without knowledge of previously defined patterns is an interesting feature of *SOFM* auto-associative neural models [9,10,11]. This situation may occur during production processes in agriculture, for example, when identifying classes of maturity of composted material.

If attribute representing classes of compost maturity have not been determined earlier, a ‘no pattern’ neural classifier seems to be the right instrument to assess compost quality. The analysis of the problem in available source material, initial research and consultation of experts showed that there is no effective and objective method of assessment of compost quality. The hybrid techniques used so far subjectively assess the colour, granulation, and smell of maturating organic matter. They usually entail costly laboratory investigations. Therefore, it is necessary to devise a new, effective, cheap, and user-friendly technique of assessment of compost maturity. 

The aim of this study was to create an *SOFM* neural model identifying quality classes of compost maturity based on information encoded in digital images. 

## 2. Materials and Methods

### 2.1. Materials

Scientific research was done in laboratories of the Institute of Biosystems Engineering, Poznań University of Life Sciences, Poland. High quality composted organic material for analyses was prepared at the Ecotechnology Laboratory. Next, images of compost samples were acquired at the Image Analysis Laboratory. 

Laboratory investigations enabled control of the composting process. A balanced mixture of waste was prepared so as to ensure optimal microbiological conditions. Thus, the duration of the composting process was reduced. The research material came from a bioreactor (scheme: Figure 1), which simulated real conditions inside a compost heap [13]. The temperature inside the bioreactor and the concentration of exhaust gases were monitored to precisely select the right moment to collect samples of the compost mixture from the bioreactor for analysis. 

Empirical data were acquired during research conducted between 2015 and 2017. The assumption of the experiment was to use waste matter sediment as a basic substrate supplemented with structural material to impart porosity, reduce humidity, and increase the content of carbon in the mixture. Following the data gathered in earlier research, easily available organic materials were chosen: wheat straw (texture was imparted, the carbon content was increased) and sawdust (the carbon content was increased, whereas humidity was reduced). The physicochemical parameters of the substrates were determined at the main phase. Thus, the percentage of the substrates in the composted mixture was estimated. The top priority was to determine the proportions to optimize the composting process, namely a carbon-to-nitrogen ratio (C:N) of 15–25, and humidity of 40–60%. 

A lightbox was designed and constructed to photograph compost at different phases of maturation and obtain a collection of digital images free of redundant artefacts (Figure 2). Shadows were eliminated by an original system of lighting the object inside the box. The lightbox was equipped with a fluorescent lamp emitting white light, ultraviolet and a mixture of both. The lightbox was integrated with a test station equipped with an automated image data acquisition system. 

The digital image acquisition apparatus was composed of the following items: − a lightbox − a steering module (a module driver connected with a step motor), − a reflex camera (*NIKON* D5100—DX matrix standard),− an *AF-S DX* 18–105 mm f/3.5–5.6G *VR Nikkor* lens and an *ARKAS TS*01 stand

The camera was set so as to take each picture in analogous lighting conditions, at a constant. Distance of 0.4 m. Thus, the natural geometrical position of the compost structure was retained in photographs. The JPEG standard was used in all photographs. 30 parameters characterising the hue of compost, which were then considered to be input variables of the *ANN* simulator, were selected as representative characteristics for the identification of compost quality classes. In accordance with the assumption, they were to characterise changes occurring in organic matter during the composting process. The following 30 descriptors were selected: median, mean value and standard variations of saturation, red colour, green colour, and blue colour luminance as well as the above parameters without the black colour.

Each of the aforementioned input variables can be presented in the *Full data* mode, which enables collection of statistical data from the picture within the RGB. Thus, a training formula consisting of 3,048 cases and defined by 30 input variables was created. The creation of the training formula was automated by means of the original *PIAO* version 3 beta IT system (Figure 2). 

### 2.2. Methods

Individual classes of compost maturation were identified with the *SOFM* neural classifier, which was generated by means of a specialized *Artificial Neural Networks* module, implemented in commercial software *Statistica ver.* 8.1. An *SOFM* neural network typically consists of an input layer and a square output layer, in which processed data (on input) are presented [15,16]. 

*SOFM* neural models are based on topological properties of the cerebral cortex. The postprocessing transformation of the output values resulted in the output variable, which had nominal nature [17]. The *SOFM* network has the form of a two-dimensional grid, with radial neurons at nodes. Each of the values represents a single class with corresponding neurons located in the output layer of the network. These neurons are characterised by the highest level of activation, which demonstrates the maximum conformity of the weight vector and the vector presented in the input pattern network. This structure showed the input layer of the *SOFM* network in a two-dimensional topological map, modelling multidimensional collection of input data. 

Due to the specific nature of *SOFM* neural networks, the training process is different than in the classic optimization of pattern balance of neural networks. The iterated training algorithm, known as the Kohonen algorithm [6], is in fact an adapted and modified version (for the needs of *SOFM* modelling networks) of the well-known k-means method. This parametric algorithm consists of 2 stages: first the centre of concentration is determined, and then the ray of proximity of specific classes is corrected (minimized) in an iterative way. This means, that the optimizing procedure of the *SOFM* network consists of the two distinctly separated phases. The first depends on proper organisation of neurons, while the other is used to precisely determine the values of balances. 

*Learned Vector Quantization (LVQ)* is a controlled version of the Kohonen algorithm [7]. The standardized Kohonen algorithm iteratively matches the locations of pattern vectors, which are stored in the radial layer of the *SOFM* network. It examines both the positions of existing vectors and training data. In fact, it is this algorithm that tries to transfer pattern vectors to the positions, which correspond to the centres of concentration occurring in data. In order to achieve the quality of classification, it was desirable for pattern vectors to be arranged within the range of classes so that they could represent natural concentrations inside each of the classes [17,18].

The generated training file, including the chosen representative features in its structure, was then used to create an *SOFM* neural classifier. The structure of the training file consisted of 30 input variables describing the colour of compost. The traits which were selected to characterise the colour of compost indicated random changes occurring in the composted organic matter. These values were obtained by means of the *PIAO* ver. 3 beta system. The selected image parameters represented saturation, luminance, red, blue, and green data. The created *PIAO* ver 3. beta system also enabled the acquisition of information on the average value, median and standard deviation of these parameters. Therefore, they were representative independent variables. The column including the numbers of compost maturation classes did not participate in the balance optimisation process (learning without supervision) and was only used to label 5 quality classes of compost of the generated topological map. The following 5 quality conditions of compost were identified (see Figure 3).

The created file included 3048 cases, which were conventionally divided into a training subset, a validating one and a testing one at a ratio of 2:1:1, respectively [19,20]. Figure 4 shows the structure of the training file. 

Figure 5 shows the procedure applied to identify neurons of compost quality classes.

In order to generate an *SOFM* neural classifier with the *LVQ* algorithm, the *Statistica Neural Networks* was applied. It is an efficient simulating device generating optimal *ANN* topology based on existing empirical data. The resulting set of networks was verified qualitatively and then the model with the classifying capability was selected. 

*Root Mean Square Error (RMSE)* is the most commonly used measure of neural network performance [21]. It is a total error made by the *ANN* on a certain set of data, determined by summing the squares of individual errors, dividing the obtained sum by the number of included values and determining the square root of the obtained quotient. The *RMSE* is usually the most convenient single interpretation value to describe the network error. The determination algorithm of the *RMSE* was implemented in in commercial software *Statistica ver.* 8.1 (TIBCO Software Inc., Palo Alto, CA, USA).

## 3. Results and Discussion

The generated topological map was optimized with the *LVQ* algorithm implemented in the *Statistica* package version 8.1. In order to carry out the training process, the *LVQ*1 algorithm, comprising a variant of the *LVQ* method was selected. It brings a pattern vector closer to a given case, if it has a class corresponding to this case, and moves it away if there are no compatible classes. The following training parameters were selected: − the number of epochs: 3000, − training rate: 0.9–0.01,− epsilon: 0.35,− beta: 0.25.

During the research a ‘mixed case’ option was added to ensure random administration of cases in the network. It also ensured that the quality of the network in each epoch based on the validating set was checked. The created and labelled *SOFM* topology map had a square structure consisting of 225 nodes (Figure 6). 

When classifying networks, quality is defined contractually with the percentage of accordant classifications. The network generated in this study reached a quality level of 0.9213, which was a very good result. The quality of 0.9213 means correct classification of 92.13% of cases. For classification networks, quality is conventionally defined by the percentage of compatible classifications. 

The *RMSE* amounted to:− 0.15324 for the training file,− 0.14722 for the validating file,− 0.14426 for the testing file.

Both the quality level and low *RMSE* values proved very good classifying properties of the neural model. The *SOFM* neural model, which was trained with the *LVQ* algorithm made without pattern classification of the degree of compost maturation, identified five quality classes. Standard labelling of the generated topological map as well as the imparting of colours to individual classes are shown in Figure 4 and Figure 5.

## 4. Conclusions

The non-pattern *SOFM* neural modelling and methods of digital image analysis identified the compost maturation classes and supported the decision-making processes during compost production. The *SOFM* network with the output variable in the form of a 15 × 15 topological map successfully and graphically identified five classes of the composted material. The analysis showed that it was enough to obtain information about the colour of composted matter to identify its maturity. The proposed method is of utilitarian significance because it can be used for automated assessment of the degree of maturity of organic material being composted and as a tool supporting the decision-making process during production. The research led to the following conclusions:

The *SOFM* network model optimized with the *LVQ*1 algorithm was characterised by the best classification parameters—its *RMS* error amounted to 0.15324 for the training file, 0.14722 for the validating file, and 0.14426 for the testing file.Optically recorded phases of compost maturation can be categorized into five classes.The *SOFM* structure with 30 input variables and 1 output variable in the form of a labelled, square topological map including 225 nodes was an optimal separating structure for five compost maturity classes, which represented five time phases.The ‘non-pattern’ *SOFM* classifier proved to be an effective instrument supporting the automated identification of five compost quality classes based on graphic information encoded in the form of digital images of the organic compost material. 

## Figures and Tables

**Figure 1 ijerph-16-03294-f001:**
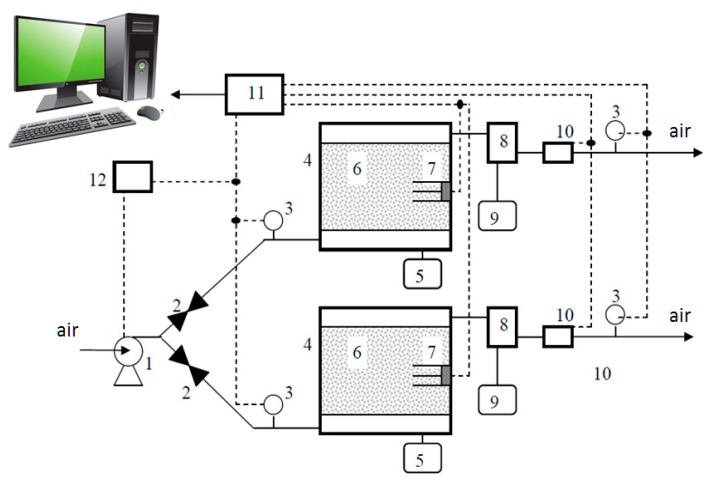
A diagram of a bioreactor used at the Ecotechnology Laboratory: 1—pump, 2—flow adjuster, 3—flow meter, 4—insulated chamber, 5—reflux tank, 6—compost mass, 7—temperature sensor complex, 8—air radiator, 9—condensate tank, 10—column with gas sensors, 11—sixteen-track recorder, 12—pump steering system [12,13].

**Figure 2 ijerph-16-03294-f002:**
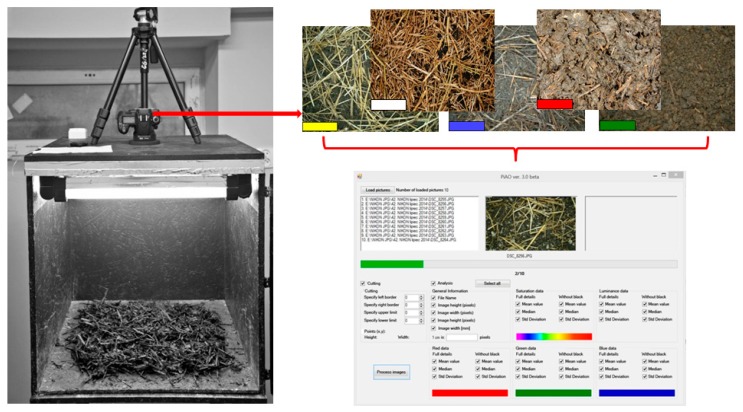
The structure of a light box for acquisition of digital images at the Image Analysis Laboratory and the mechanism of creation of a training set with the *PIAO* IT system ver.3 beta [14].

**Figure 3 ijerph-16-03294-f003:**
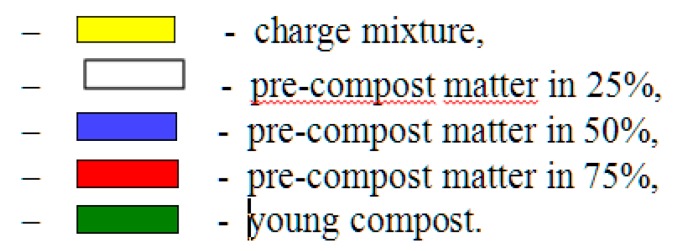
Quality conditions of the compost.

**Figure 4 ijerph-16-03294-f004:**
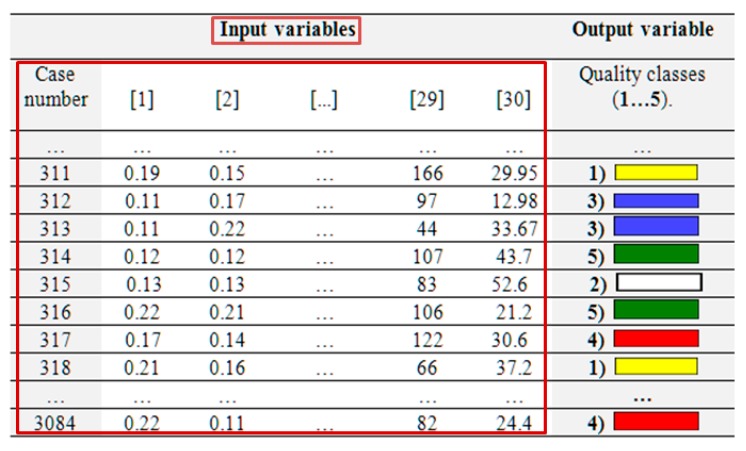
A fragment of the training file.

**Figure 5 ijerph-16-03294-f005:**
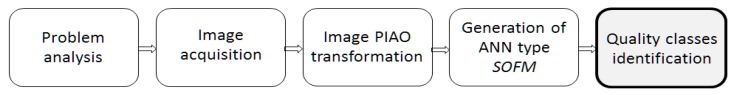
The pattern of creation of a *Self-Organising Feature Map (SOFM)* neural classifier.

**Figure 6 ijerph-16-03294-f006:**
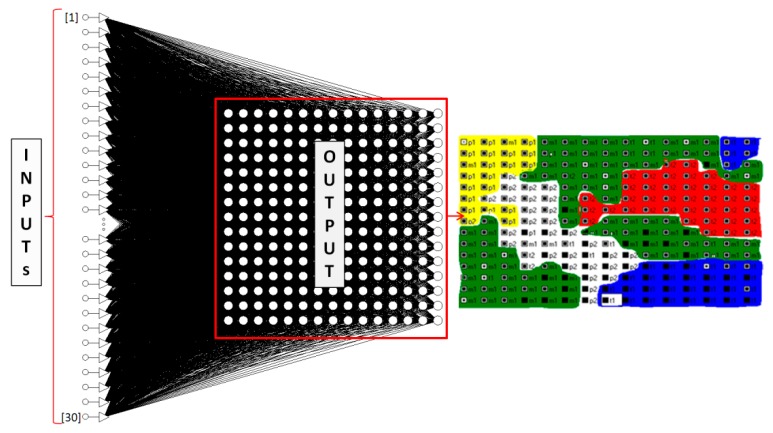
The illustrative nature of the structure of the optimal *SOFM* neural model.

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
