# Peer review of "Neural Classification of Compost Maturity by Means of the Self-Organising Feature Map Artificial Neural Network and Learning Vector Quantization Algorithm"

_ijerph, 2019, doi:10.3390/ijerph16183294_

Round 1

Reviewer 1 Report

The comments and suggestion for authors are placed in the attached file.

Author Response

Poznań, 27 August 2019

Dear Reviewer 1,

Thank you so much for your valuable comments and suggestions.

Changes in the text - introduced by the authors and native speaker - are marked in red and green. Green color is primarily related to the language corrections.

In my opinion, the title of the article is somewhat misleading and ambiguous. After all, NN SOM is implemented in the form of an algorithm just like LVQ, so what's the difference between them semantic?

Learned Vector Quantization provides a supervised version of the Kohonen training algorithm.  LVQ is a controlled version of the Kohonen learning algorithm. The standard Kohonen algorithm iteratively adjusts the position of the model vectors stored in the radial layer of the Kohonen network because it considers only the positions of existing vectors and training data.  In fact, the algorithm tries to move the model vectors to the positions corresponding to the cluster centres in the training data.  However, the class labels of the training data cases are not taken into account.  To achieve the best classification quality it is desirable to position model vectors within the range of classes to some extent so that they will represent natural clusters in each class.  A vector located on the border between the classes, at an equal distance from cases of both classes, is not much use for distinguishing the class.  On the other hand, vectors located within class boundaries may be very useful.

There are several variants of the LVQ algorithm. The Statistica Neural Networks supports three of them. The basic version, LVQ1, is very similar to the Kohonen training algorithm.  The closest vector to the training case is selected and its position is modified.  However, while the Kohonen algorithm would move this vector towards the training case, LVQ1 checks whether the class label of the vector is the same as that of the training case.  If it is so, the model vector is moved towards the training case; if not, the vector is moved away from the case. 

Authors presented the use of self-organizing neural networks of the standard Kohonen map and its LVQ version enabling labeling of classes extracted from the training set in the learning stage. The authors declared that the main purpose of the article is "to create neural model type SOFM identifying quality classes of compost maturity in the process of its maturation made on the basis of information encoded in the graphic form in the form of digital pictures". While the main purpose of the presented researches were clearly defined, their scope is unclear, as well as the manner of their implementation. Having regard to the above, I kindly invite authors to answer the following questions:

What does shadow-free chamber mean? Is it possible to obtain a flat 2D image free of shadows by illuminating a 3D object with the complex geometry of its surface? In my opinion, this term is not entirely true.

Original lightboxes were made for the experiment. Cubical boxes sized 500 × 500 × 500 mm were built from wood-based composite OSB (Oriented Strand Board). Black paint on the internal walls absorbed light and prevented reflections.

What effect does image acquisition in JPEG format have? Do the authors want to emphasize the effectiveness of the demonstrated method regardless of the degree of compression of image information implemented by the JPEG algorithm, or was the recording carried out in a given compression? If so, in which one?

JPEG is a common standard. The authors used it to easily obtain images with the proposed method. JPEG compression also enabled the authors to use of large size images (9 million pixels) for tests.

Why did the authors adopt color [0,0,0] elimination? Is not the presence of black color a carrier of information on the structure of compost, and thus the state of its maturity?

Corrected The text from lines 126-127 has been removed.

How did the authors define the number of parameters characterizing the shade of compost? Why 30?

The selection of 30 representative parameters was preceded by a series of simulation tests. 300 parameters characterizing the colour were analysed. The sensitivity of the generated neural models to the input parameters was analysed with a standard method. The number of input variables was reduced to 30 representative parameters.

Have other sets of such parameters been considered? How was the learning set of 3048 vectors obtained?

The training set was obtained by the acquisition and analysis of 10,000 digital photos of compost at various stages of maturity, which were taken in a laboratory.

Was the standardization necessary for them to ensure a convergent learning process for SOM networks? What learning algorithm was used at work, WTA?

The LVQ algorithm was used to generate the SOFM network, which is a controlled version of the Kohonen training algorithm implemented in the Statistica 8.1 package. As indicated in the title, the LVQ1 algorithm, which is a version of the LVQ algorithm, was used in the study. The basic LVQ1 version is very similar to the Kohonen training algorithm, where the closest vector to a given learning case is searched and its location is modified accordingly. However, while the Kohonen algorithm shifts the vector towards the learning case, LVQ1 checks the compatibility of the vector and case classes. The LVQ1 algorithm approximates the model vector to a given case provided it has the same class as the case, or moves it away in case of class incompatibility. Standard procedures included in the Statistica 8.1 software were used to create a Kohonen network with the LVQ1 algorithm.

What does it mean and what is the role of process automation implemented by the PIAO system?

The PIAO ver. 3 beta program is an original, dedicated IT tool for the generation of training sets of csv. files, as this format is compatible with the artificial neural network simulator used in the study. Training files are created automatically based on JPEG images. This facilitates and speeds up the ANN simulation process.

What is the meaning of indexes of accepted parameters 1 to 30 have for the reader of the work? Does their order affect the learning process?

Parameter indexes 1-30 are only symbolic to reduce the entries in Table 1 (like the entries for classes 1-5). The order is of no significance to the training set structure.

What are their physical interpretations? Why in the text of the manuscript there are no methods of obtaining them given, although they are crucial, e.g. to repeat or verify the described experiments?

30 traits referring to the colour of the compost were selected, as they characterized random changes occurring in the organic matter being composted. These values were obtained with the PIAO ver. 3 beta system. The selected image parameters represented data on saturation, luminance, red, blue, and green colours. The PIAO ver 3 beta IT system also enabled the acquisition of information on the average value, median and standard deviation of these parameters. Thus, these were representative independent variables.

Did the authors carry out experiments for other architectures? Why is the SOFM 10x10 map size optimal? (in line 212, the authors declare its size 100x100 ?!)

The text has been corrected.

Why is the size of the network so oversized if only 5 classes of compost material are identified? How was the 15x15 topology obtained? - right part of the drawing 5.

Simulations were carried out for 10 SOFM structures (range: 10 × 10,   11 × 11 ……  20 × 20). The Kohonen map architecture with the 15 × 15 structure was selected.

The drawing is for reference only. Correction of Fig. 5  

In the L. 209, the authors state that digital image processing methods were used. What were these methods and for what purpose were they used?

L209 (currently 219) does not contain information about digital image processing. Selected methods of image analysis were used in the study. They were not described in detail due to the limited format of the article. The necessary image processing techniques were implemented in the auxiliary PIAO ver.3 beta IT system.

As you know, a key element of the effective learning process of the Kohonen network type is controlling the dynamics of the learning process, which is crucial for the proper discovery of classes and their further proper identification. Is the function of changing the learning rate coefficient known during the learning process? If so, then why not put it in the text of manuscript?

The notes do not apply to training with the LVQ algorithm.

The text has been corrected.

Is it possible to formulate objective conclusions and generalize the measurement method L.219-L227 if all experiments were carried out for one set of parameters? If other attempts were made (I am convinced that it was so), it is worth quoting them here or earlier, pointing to the positive and negative elements of choosing the size of the network the strategy of preprocessing and post processing of experiment data.

For comparative purposes, the authors tested neural classifiers with different topologies (MLP, RGB, PNN). However, these networks require the ‘teacher’ training technique, which implies the training set structure. This requires information about the number of classes. The idea of the study was to identify the number of compost classes. Kohonen topology was used for this purpose, as it does not involve  a ‘teacher’.

It is worth for the reader include Pearson distribution charts including the training, validation and test file, which in graphic form will allow for easy and quick assessment of the proper functioning of the neural network.

The Pearson correlation coefficient is a standard instrument for assessing the quality of ANNs, which solve regression problems (ANN regression). The SOFM ANN was tested in the study as a classification tool (ANN classification).

Formatting:

The authors have corrected the article according to the comments.

Specific comments:

Please check the following list to avoid some editorial ambiguities.

L16 “y”? L46 the SOM and SOFM are used interchangeably without explanation why? and for what? L67 –Materials L149 unexpected end of the line L157 – the sentence “teaching data” should be replaced by more suitable training set or

learning data.

Reviewer 2 Report

Abstract not clearly. The abstract is too general, especially at the end where the main results are summarized The authors have improved the manuscript, but the description of the methodology is confusing and very hard to understand. Artificial Neural Network Type Self Organizing Feature Map and Algorithm learning vector quantization not clearly, must be explain and how input and process The result have to compare with another method

Author Response

Poznań, 27 August 2019

Dear Reviewer 2,

Thank you so much for your valuable comments and suggestions.

Changes in the text - introduced by the authors and native speaker - are marked in red and green. Green color is primarily related to the language corrections.

Abstract not clearly. The abstract is too general, especially at the end where the main results are summarized.

The research resulted in the development of original software for quick and easy assessment of compost maturity. The generated SOFM neural model was the kernel of the constructed IT system. The article does not describe the new software because it will be presented in a separate publication.

The authors have improved the manuscript, but the description of the methodology is confusing and very hard to understand.

The text has been supplemented.

Artificial Neural Network Type Self Organizing Feature Map and Algorithm learning vector quantization not clearly, must be explain and how input and process The result have to compare with another method.

The SOFM neural model with the LVQ1 algorithm underwent standard qualitative verification, using the procedures included in the Statistica Neural Networks module implemented in the Statistica ver. 8.1 package.

Kind regards,

Agnieszka Pilarska

Round 2

Reviewer 2 Report

How do you get value in page 6 line 180 to 184. Explain it

Author Response

Response to the Reviewers’ Comments

Poznań, 04-09-2019

Dear Reviewer,

Thank you for your valuable comments and kindness. I hope this answer is acceptable to you.The text has been supplemented.

For classification networks, quality is conventionally defined by the percentage of compatible classifications. Record: quality of 0.9213 means correct classification of 92.13% of cases.

RMSE (Root Mean Square Error) is the most commonly used measure of neural network performance. It is a total error made by the ANN on a certain set of data (teaching, testing or validation), determined by summing the squares of individual errors, dividing the obtained sum by the number of included values ​​and determining the square root of the obtained quotient. The RMSE is usually the most convenient single interpretation value to describe the network error.

The algorithm for determining the RMSE separately for the data (teaching, testing and validation) is implemented in the Statistica package ver. 8.1. It is a standard procedure for identifying the error made by ANN and allows for a qualitative analysis of the neural classifier.

Kind regards,

Agnieszka Pilarska

on behalf of the Authors
